# Use of Advance Directives in US Veterans and Non-Veterans: Findings from the Decedents of the Health and Retirement Study 1992–2014

**DOI:** 10.3390/healthcare11131824

**Published:** 2023-06-22

**Authors:** Ho-Jui Tung, Ming-Chin Yeh

**Affiliations:** 1Department of Health Policy and Community Health, Jiann-Ping Hsu College of Public Health, Georgia Southern University, P.O. Box 8015, Statesboro, GA 30460-8015, USA; 2Nutrition Program, Hunter College, City University of New York, New York, NY 10065, USA

**Keywords:** veterans, living will, durable power of attorney for healthcare, advance directives, advanced care planning, Veterans Health Administration

## Abstract

Evidence shows that older patients with advance directives such as a living will, or durable power of attorney for healthcare, are more likely to receive care consistent with their preferences at the end of life. Less is known about the use of advance directives between veteran and non-veteran older Americans. Using data from the decedents of a longitudinal survey, we explore whether there is a difference in having an established advance directive between the veteran and non-veteran decedents. Data were taken from the Harmonized End of Life data sets, a linked collection of variables derived from the Health and Retirement Study (HRS) Exit Interview. Only male decedents were included in the current analysis (N = 4828). The dependent variable, having an established advance directive, was measured by asking the proxy, “whether the deceased respondent ever provided written instructions about the treatment or care he/she wanted to receive during the final days of his/her life” and “whether the deceased respondent had a Durable Power of Attorney for healthcare?” A “yes” to either of the two items was counted as having an advance directive. The independent variable, veteran status, was determined by asking participants, “Have you ever served in the active military of the United States?” at their first HRS core interview. Logistic regression was used to predict the likelihood of having an established advance directive. While there was no difference in having an advance directive between male veteran and non-veteran decedents during the earlier follow-up period (from 1992 to 2003), male veterans who died during the second half of the study period (from 2004 to 2014) were more likely to have an established advance directive than their non-veteran counterparts (OR = 1.24, *p* < 0.05). Other factors positively associated with having an established advance directive include dying at older ages, higher educational attainment, needing assistance in activities of daily living and being bedridden three months before death, while Black decedents and those who were married were less likely to have an advance directive in place. Our findings suggest male veterans were more likely to have an established advance directive, an indicator for better end-of-life care, than their non-veteran counterparts. This observed difference coincides with a time when the Veterans Health Administration (VHA) increased its investment in end-of-life care. More studies are needed to confirm if this higher utilization of advance directives and care planning among veterans can be attributed to the improved access and quality of end-of-life care in the VHA system.

## 1. Introduction

Advances in medical technologies (e.g., chemotherapy, tube-feeding, and ventilators) have made dying an increasingly prolonged and “medicalized” process [1]. In many cases, these life-sustaining treatments may prolong the lives of patients with terminal illnesses, but not necessarily enhance the quality of their lives. Dying patients may go through invasive, costly, and futile interventions in acute care settings [2] that are incongruent with the characteristics of quality end-of-life care both patients and families prefer [3]. Advance directives are care planning tools, introduced as a solution to improve the quality of end-of-life care [4] and they can take the form of a living will or durable power of attorney for healthcare (DPAHC). Patients can document their treatment preferences, along with a variety of end-of-life choices, when they are still capable of making decisions.

Evidence indicates that advance care planning (ACP) is less likely to take place in hospitals and intensive care units, where timely transitions to palliative care could be slowed and a patient’s preferences might not be honored sometimes [5]. Although there are reports on the limitations, regarding the efficacy and effectiveness of advance directives in clinical practices [1,6]. Studies have found that the use of ACP is associated with better transitions to palliative care and more use of hospice where patient-centered care is honored [7,8,9]. Considered an important indicator of better planning for end-of-life care, several systematic review articles also conclude that the use of ACP is associated with other positive outcomes, such as dying in preferred place, reducing invasive and futile treatments, and achieving some characteristics of “good death” [10,11,12,13]. It has been recognized that advance directives are not a panacea to all the problems in end-of-life care. Instead, they should serve as the starting point for ongoing communication between patients, surrogates, and health professionals [14].

Empirical research on the use of ACP has identified a variety of factors associated with the use of ACP, including age, race/ethnicity, socioeconomic status (SES), psychological, religious, and attitudinal factors [1]. Racial and ethnic disparities have been heavily investigated and significant racial disparities in the use of ACP are identified [15,16], where African Americans are less likely to have an established advance directive than their white counterparts. Significant differences in the use of advance directives are also found among patients with different terminal illnesses [11]. Patients who die of different diseases may go through different trajectories of terminal decline [17], which could lead to multiple transitions between different care settings near the end of life. However, little is known about the use of advance directives and care planning at the end of life between veteran and non-veteran older adults in the United States. 

For male Americans who were born before the 1940s, veteran status is an important mediator of their aging experiences, because many of them had been drafted into the military and a large proportion of these veterans had served in World War II, the Korean War, or the Vietnam War. A theoretical perspective in social gerontology, the life course perspective, highlights the link between early experiences and later developments in human lives. In a similar vein, the lifespan view of military influences on aging US veterans also asserts that the effects of military service are lifelong [18]. Veterans usually form strong network ties through their military experiences and people’s social networks provide a structure where potential resources are embedded [19,20]. The literature of healthcare utilization and the social process of help-seeking behavior have suggested that an individual’s network is important in channeling their entrance into care [21].

Furthermore, as the largest healthcare delivery system in the country, the Veterans Health Administration (VHA) is charged with the provision of healthcare to qualified US veterans. The increasing number of aging veterans has been an important driving force for the VHA to take the lead in developing end-of-life care programs and initiatives (e.g., hospice and palliative care) [22,23]. There have been reports on the VHA’s investment in improving the access and quality of end-of-life care since the 1990s [5,24,25]. Several policy evaluation studies also found that these programs have improved the quality, availability, and accessibility of end-of-life care in the VHA system [23,26]. However, most of these studies analyzed data from the medical records kept within the VHA system, so that their samples consisted mainly of the veterans covered by the VHA. In this study, we used data from a nationally representative survey, in which both veterans and non-veteran older adults were sampled. We compared the difference in the use of advance directives (either a living will or DPAHC) between the deceased veterans and non-veterans from a longitudinal survey over two decades.

## 2. Methods

### 2.1. Data and Samples

Launched in 1992, the Health and Retirement Study (HRS) is a panel-designed longitudinal survey of those over the age of 50 in the United States [27]. The survey draws a multistage probability sample that is nationally representative of the U.S. population. Using both face-to-face and telephone interviews, the survey collects information addressing many important questions related to the aging experiences in America. Core interviews were conducted every two years and, for the deceased HRS participants, exit interviews were arranged with their proxies to seek information about the decedents, including the diseases and causes of deaths, healthcare utilization, and end-of-life planning.

Data for the current study were taken from the Harmonized HRS End of Life data files, a streamlined collection of variables derived from the first HRS Exit Interviews from 1994 to 2014 [28]. A total of 12,952 HRS participants had died between 1992 and 2014. However, 2623 of the decedents whose proxies did not provide information on advance directives. Plus, for these cohorts of older Americans, military service was predominantly a male role, we further excluded all of the 5471 female decedents from the analyzed sample. After excluding another 30 cases with missing values on other variables, a total of 4828 male decedents were available for the current analysis. 

### 2.2. Measures

For the dependent variable, the provision of advance directives was measured by two questions: “whether the deceased respondent ever provided written instructions about the treatment or care he/she wanted to receive during the final days of his/her life?” and “whether the deceased respondent had a Durable Power of Attorney for health care?” If the answers to either of the two items were “yes”, the dependent variable was coded 1 (having an established advance directive). It was coded 0 if both answers were “no”, indicating that the decedent had neither of the two types of advance directive.

For the independent variables, age at death was treated as a continuous measure. Over 96 percent of the male decedents were either white or Black, so race was dichotomized into Black (=1) and all others (=0). Educational attainment was measured as years of schooling and this measure was treated as a continuous variable. Marital status at death was also a dichotomous measure. The proxy was asked if the respondent was married (or partnered) at the time of death (yes = 1; otherwise = 0).

Caring for dying patients can be both physically and emotionally challenging for their family caregivers. Two indicators of care burden were also included to predict the dependent variable, the likelihood of having an established advance directive before death. In the survey, proxies were asked whether a spouse, child/grandchild, or other relatives had helped with Activity of Daily Living (ADLs) in the three months prior to death. A “yes” to any of the six ADLs (eating, bathing, dressing, walking, toileting, and getting in and out of bed) was coded as 1 and 0 for otherwise. Bedridden status was coded as 1 and 0 for otherwise if the respondent spent more than half the day in bed over 85 days during the last three months before death. These measures could be indications of an expected death and a prompt for end-of-life care planning activities. Since the deaths in our sample occurred over a long period, the social norms regarding the use of advance directives and the availability of care-planning tools could change considerably over time. Thus, a dichotomous variable indicating whether the death occurred during the earlier half (from 1992 and 2003) or the latter half (from 2004 to 2014) of the study period was also included. Finally, the prognosis of patients dying of cancer is more predictable and consistent when compared to other common causes of mortality [29]. Decedents whose main cause of death was cancer were singled out as a predictor for the use of advance directives. The main cause of death in the survey was determined by asking the proxies the following open-ended question, “What was the major illness that led to (her/his) death?”. The reported illnesses were then recoded according to the Health Conditions Master Code developed by the HRS [28]. If the main cause of death was cancer, it was coded 1. For all other causes of death, it was coded 0. 

### 2.3. Analysis

Binary logistic regression models, where the dependent variable was constructed as the probability of having an advance directive in place versus not having an advance directive. The main independent variable, veteran status, was used to predict the likelihood of having an advance directive, while controlling for other covariates. Odds ratios (and their 95 percent of confidence intervals) were used to evaluate the significance of the included predictors. Separated analyses were performed for decedents whose death occurred during the earlier half (from 1992 to 2003) and those who died during the latter half of the study period (from 2004 to 2014).

## 3. Results

Among the 4828 male decedents, about 58 percent of them (2783 out of 4828) were identified as veterans. The high percentage of veterans in our sample is because over 90 percent of the male decedents in our sample were born before 1941. For this cohort of male Americans, a military conscription was in place until 1973. From the descriptive statistics presented in Table 1, we know that, on average, veterans died older and had more years of schooling than those of the non-veteran decedents. The percentage of African Americans was significantly lower among the veteran group when compared to the non-veterans. The percentages of having a living will or a durable power of attorney for healthcare were also significantly higher among the veterans, when compared to their non-veteran counterparts. 

Table 2 presents the logistic regression results for the whole sample. Veterans were more likely to have an established advance directive (odds ratio, OR = 1.15, *p* < 0.05) than that of the non-veterans, after controlling for other covariates. Deceased HRS participants who died at an older age (OR = 1.04, *p* < 0.001) and who had more years of schooling (OR = 1.12, *p* < 0.001) were more likely to have an established advance directive. Decedents who needed ADL assistance three months before death, those who were bedridden three months before death, and those who died during the latter half of the study period, had a significantly higher chance of having an advance directive. On the other hand, African Americans and those who were married at the time of death were less likely to have an advance directive in place.

Table 3 presents separated logistic regression models in predicting the use of advance directives for decedents who died earlier (from 19992 to 2003) and whose deaths occurred later in time (from 2004 to 2014). We found that there was no significant difference in having an established advance directive between male veteran and non-veteran decedents whose death occurred during the earlier-half study period. The significant difference between veterans and non-veterans happened to concentrate on decedents who died during the latter half of the study period (OR = 1.24, *p* = 0.02). For other covariates, the association patterns stayed the same. 

## 4. Discussion

As a tool to facilitate decision-making and communication at the end of life, ACP is associated with better end-of-life care, such as dying in preferred place and healthcare cost savings [12]. Furthermore, having an advance directive and ACP at the end of life are also associated with a reduced decision-making burden and improved well-being for dying patients’ family members [30]. In this study, we used data collected from both the proxies and the deceased participants of a longitudinal survey to compare the rates of having an established advance directive between male veterans and non-veterans. The results show that male veterans had a significantly higher use of advance directives, an indicator of quality end-of-life care, than their non-veteran counterparts during the latter half of the study period (from 2004 to 2014). We also found that, regardless of their veteran status, male decedents, who died at older ages and who had higher educational attainment, were more likely to have an established advance directive. African American decedents and those who were married at the time of death were less likely to have an advance directive in place. Decedents with a higher care burden (those who needed ADL assistance and were bedridden three months before death) were also more likely to have an established advance directive.

The percentage of older adults with a written advance directive has been rising since the Patient Self-determination Act was passed in 1990 [1]. When asked, a great majority of Americans believe that having a family conversation about their wishes regarding life-sustaining treatments at the end of life are important, but much lower percentages of people have done so [5]. Conducted on adults of various ages, surveys on ACP activities indicated that some 23 to 54 percent Americans had a written advance directive in place and the rates were as high as 70 percent among older adults who had a terminal illness [1,5,9]. In the current analysis, the percentages of having either a living will or a DPAHC among the decedents who died during the first half of the study period were 47.8 percent for non-veterans and 54.2 percent for veterans. By the latter half of the study period (from 2004 to 2014), the rate difference had increased to 54.8 percent for non-veterans and 69.2 percent for veterans. 

We are not completely clear about all the factors contributing to the observed difference in the use of advance directives between veteran and non-veteran HRS decedents. However, our findings indicate that the difference was significant only in the latter half of the study period (from 2004 to 2014). This time frame overlaps with a period when several end-of-life care programs and initiatives were launched and implemented by the VHA system. For example, the VHA was the first to require hospice consultation teams to be established in all its care facilities in 1992 and the Hospice–Veteran Partnership Program (launched in 2001) made hospice and palliative care widely available to veterans and their caregivers [22,24,25]. The Bereaved Family Survey was also launched in 2008 by the VHA to evaluate performance and monitor family members’ perceptions of veterans’ end-of-life care [29,31].

Moreover, for non-veteran older Americans covered by Medicare (the federal health insurance program for older people aged 65 or older in the United States), they must waive aggressive treatments in order to be eligible for hospice care. In contrast to Medicare beneficiaries, the VHA allows for the provision of concurrent care while the patient is in hospice [32,33]. Thanks to veterans’ sacrifice to the country, these generous care benefits often receive bipartisan support in Congress. It is reasonable to speculate that these programs and initiatives have significantly improved the quality of end-of-life care and access to care planning tools within the VHA system.

There might be another organizational advantage of the VHA system in carrying through its end-of-life care policies. The VHA is the largest integrated healthcare delivery system in the United States. Different from Medicare, which functions as a healthcare purchaser, the VHA provides healthcare directly to qualified veterans. Over the past decades, Medicare has also expanded its coverage for end-of-life counseling on advance directives and hospice use [1,34]. However, as a care purchaser, Medicare can only implement its policy initiatives through incentives in reimbursing contracted care providers and managed-care organizations. The centralized VHA system with a salaried medical staff makes coordination more likely to occur, so it would be easier to put established policies into place.

Finally, it should be noted that there are several limitations in our study. First, this study focusses mostly on the birth cohorts of male HRS participants who lived through a time when military conscription was instituted in the United States. Military drafting was ended when the All-Volunteer Force (AVF) policy was established in 1973. Additionally, the centralized VHA healthcare system is supervised by the Department of Veteran Affairs, a US cabinet-level agency under the executive branch. These cohort-historic factors are culture- and country-specific, so the findings and implications of this study may not be applicable internationally. 

Second, many veterans are eligible for both VHA care and Medicare. It is possible that some veterans could seek care in a non-VHA facility, meaning they would not be exposed to the organizational advantages and end-of-life care benefits provided by the VHA system. A survey in 2010 found that 77 percent of the veterans enrolled in the VHA were eligible for additional healthcare coverage and VHA care users were more likely to be older [35]. However, over 90 percent of the male veterans in our sample were born before 1941 (the pre-Vietnam-era veteran) [36]. It is reasonable to believe that, when presented with multiple options for end-of-life care, most veterans would seek VHA care, where better end-of-life benefits were offered. 

Third, the HRS exit surveys were conducted by interviewing the proxy informants, so information reported by the proxies is subject to recall biases. Plus, not all the proxies for the HRS decedents knew all the details of end-of-life care planning pertaining to the deceased participants. About 20.3 percent of the decedents in the Harmonized HRS End of Life data (out of the 12,952 deaths of HRS participants recorded between 1992 and 2014) had missing information on the details of advance directives. It is possible that potential biases could be introduced due to the missing observations. Fortunately, the variables included in this study were mostly restricted to observable behaviors and facts of the deceased respondents reported by their proxies. According to the data description documents, about 95 percent of the proxies interviewed in the Exit Interview Surveys were related to the deceased participants [28], so reporting errors should be minimal. 

Lastly, it is still possible that some unmeasured confounders might explain the observed difference between veterans and non-veterans in the use of advance directives. More studies are needed to confirm whether the observed difference in the use of advance directives can be attributed to the improved access and quality end-of-life care provided by the VHA system. 

## Figures and Tables

**Table 1 healthcare-11-01824-t001:** Selected characteristics of the male decedents of Health and Retirement Study participants by veteran status, 1992 to 2014.

Predictors	Veteran (N = 2783)	Non-Veteran (N = 2045)
Mean Age at death	78.2 (9.3)	77.4 (11.8)
Years in schools	12.3 (3.1)	10.0 (4.2)
African Americans (yes)	296 (10.6)	430 (21.0)
Marital status at death (married)	1858 (66.8)	1214 (59.4)
Death occurred between 1992 and 2003 (yes)	1178 (42.3)	897 (43.9)
Cancer as the main cause of death (yes)	803 (28.9)	530 (25.9)
Needed help with any activities of daily living 3 months before death (yes)	1320 (47.4)	921 (45.0)
Bedridden 3 months before death (yes)	674 (24.2)	512 (25.0)
Had a living will (yes)	1276 (45.8)	687 (33.6)
Had a durable power of attorney (yes)	1531 (55.0)	916 (44.8)

Note: For categorical variables, the number of cases and percentage (in parentheses) are presented and for continuous variables (age and years of schooling), means and standard deviations (in parentheses) are presented. Numbers and percentages for the two types of advance directives, a living will and a durable power of attorney for healthcare (DPAHC) were presented separately here, but they were combined to form the dependent variable in the logistic regression analyses.

**Table 2 healthcare-11-01824-t002:** Odds ratios of having an established advance directive among the male decedents of the Health and Retirement Study, 1992 to 2014.

Predictors	Odds Ratio (95% Confidence Intervals)
Veteran status	
No	1.0
Yes	1.15 (1.01, 1.31) *
Age at death	1.04 (1.04, 1.05) ***
Years in schools	1.12 (1.10, 1.14) ***
African American	
No	1.0
Yes	0.36 (0.30, 0.43) ***
Marital status at death	
No	1.0
Yes	0.74 (0.65, 0.85) ***
Cancer as the main cause of death	
No	1.0
Yes	1.35 (1.17, 1.56) ***
Needed help with any activities of daily living 3 months before death	
No	1.0
Yes	2.26 (1.94, 2.62) ***
Bedridden 3 months before death	
No	1.0
Yes	2.27 (1.91, 2.69) ***
Death occurred	
Between 1992 and 2003	1.0
Between 2004 and 2014	1.45 (1.28, 1.65) ***
−2 Log Likelihood (degrees of freedom)	5743.98 (9)

Note: * *p* < 0.05, *** *p* < 0.001.

**Table 3 healthcare-11-01824-t003:** Odds ratios of having an established advance directive among male decedents of the Health and Retirement Study, separated by death period.

Predictors	Decedents Whose Death Occurred between 1992 and 2003 (N = 2075)	Decedents Whose Death Occurred between 2004 and 2014 (N = 2753)
Veteran status		
No	1.0	1.0
Yes	1.04 (0.85, 1.27)	1.24 (1.03, 1.48) *
Age at death	1.04 (1.03, 1.05) ***	1.04 (1.03, 1.05) ***
Years of schooling	1.12 (1.09, 1.15) ***	1.11 (1.08, 1.14) ***
African American		
No	1.0	1.0
Yes	0.36 (0.27, 0.47) ***	0.37 (0.29, 0.46) ***
Married at death		
No	1.0	1.0
Yes	0.77 (0.63, 0.94) *	0.73 (0.60, 0.87) **
Cancer as the main cause of death		
No	1.0	1.0
Yes	1.37 (1.11, 1.70) **	1.34 (1.10, 1.63) **
Needed help with any activities of daily living or 3 months before death		
No	1.0	1.0
Yes	2.15 (1.72, 2.69) ***	2.34 (1.91, 2.87) ***
Bedridden 3 months before death		
No	1.0	1.0
Yes	2.13 (1.66, 2.72) ***	2.41 (1.90, 3.05) ***
−2 log likelihood (degrees of freedom)	2569.43 (8)	3171.99 (8)

Note: * *p* < 0.05, ** *p* < 0.01, *** *p* < 0.001.

## Data Availability

Study data were downloaded from a publicly accessible website, the Health and Retirement Study.

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
