# Peer review of "Use of Advance Directives in US Veterans and Non-Veterans: Findings from the Decedents of the Health and Retirement Study 1992–2014"

_healthcare, 2023, doi:10.3390/healthcare11131824_

Round 1

Reviewer 1 Report

This is a well written and a relatively first in attempting to define do veterans expect or require different care at end of life. In that sense it is ready for publication. I only have a minor suggestion to the authors to be clear in the title, abstract and main body that this is a US study, as much of that described can be attributed to the veteran health care in the USA and is not transferable to other nations.

There is a minor typo that needs addressing in line 167, 1992 not 19992.

Author Response

We appreciate the reviewer’s encouraging comments. It is true that this study focusses mostly on the birth cohorts of male HRS participants who had lived through a time when military conscription was instituted in the United States. These cohort-historic factors are very country-specific, so the findings and implications of this study may be applicable internationally. We have emphasized this in the revised manuscript. We apologize for the typo.

Reviewer 2 Report

Thank you for giving me the time to review your manuscript. This manuscript is interesting for considering Use of Advance Directives between Veteran and Non-veteran Decedents. Regarding the contents, the following revision should be considered.

Title

The title should include the study design.

The abstract

-The abstract should include specific research design descriptions.

Introduction

Generally, there is no paragraph writing. The background contains few paragraphs. The author should focus on theory building, the problems, and research question paragraphs. The first and second paragraphs should focus on general information regarding advanced directives. Moreover, the third and fourth paragraphs should introduce the research question as the theoretical and conceptual framework, including which country’s conditions the authors focused on in international contexts and research questions.

-This study focuses on the conditions of USA. However, there is a lack of reasons why this research focus on the country.

-The introduction should clearly include this study's international contexts and research questions.

Method

-Why did this study focus on participants in USA? 

-Each variable included in this study should be explained specifically.

-Sample calculation should be described clearly.

-Logic regression model should be explained more clearly regarding why each variable is used in the analysis.

Result

-Regarding subscale analysis, the issue of multiple analyses present. The authors should show the p-values with Bonferroni correction.

-This section is not reader friendly and need subsections.

Discussion

With the same as the background, the authors should use paragraph writing for logical theory building. 

This study should describe the limitation of sampling bias, the results' applicability to other settings, and the future investigation in the limitation part.

In the conclusion or discussion, the study’s strengths should be focused on international readers.

Conclusion

The conclusion section describes the result again. The authors should add a description of news regarding this research field.

Author Response

Thank you for giving me the time to review your manuscript. This manuscript is interesting for considering Use of Advance Directives between Veteran and Non-veteran Decedents. Regarding the contents, the following revision should be considered.

Title: The title should include the study design.

Author’s response:

Thank you for the comment, we have adjusted the title.

The abstract: The abstract should include specific research design descriptions.

Author’s response: Thank you for the comment, we have added more descriptions to clarify the study design.

Introduction
Generally, there is no paragraph writing. The background contains few paragraphs. The author should focus on theory building, the problems, and research question paragraphs. The first and second paragraphs should focus on general information regarding advanced directives. Moreover, the third and fourth paragraphs should introduce the research question as the theoretical and conceptual framework, including which country’s conditions the authors focused on in international contexts and research questions.

Author’s response: We thank the very constructive comments from the reviewer. We have adopted the recommended paragraph writing to organize the flow of our arguments, from general information regarding advance directive to research question to theoretical perspective, in the Introduction section.

-This study focuses on the conditions of USA. However, there is a lack of reasons why this research focus on the country.

-The introduction should clearly include this study's international contexts and research questions.

Author’s response: We appreciate the reviewer’s comments and understand the reviewer’s concern. We know that the journal has an international readership. However, our study focusses mostly on the birth cohorts of male HRS participants who had lived through a time when military conscription was instituted in the United States. The VHA health care system is supervised by the Department of Veteran Affairs, a US cabinet-level agency under the executive branch. These cohort-historic factors are country-specific, so the findings and implications of this study may not be applicable internationally. We have explained why we took this approach in the revised manuscript.

Method
-Why did this study focus on participants in USA? 

Author’s response: Again, we understand the reviewer’s concerns and please see our explanations above.

-Each variable included in this study should be explained specifically.

Author’s response: We thank the reviewer’s comments. We have explained how we measured each of the variables in the Methods section. For the dependent variable, use of advance directives, has been discussed clearly in the text. There were 8 independent variables included in the logistic regression models. We did not include gender, because we focused only male decedents only. Age, level of education, and race are so fundamental, and they need to be controlled for in our models. Literature shows that marital status and is associated with advance care planning. We know that males have a shorter life expectancy than that of females and our sample contained only the male decedents, so there were considerable variations in marital status in our sample. We have explained the reason to include the variable, “cancer as the main cause of death,” because cancer is the most common diagnosis among hospice patients. For the two indicators of care burden “helping with Activity of Daily Living (ADLs) in the three months prior to death” and “being bedridden 3 months before death,” they are indications of an expected death. These indications might be associated with care planning activities.

-Sample calculation should be described clearly

Author’s response: We thank the reviewer’s suggestion. We have re-organized the description regarding the sample size evolution.  

-Logic regression model should be explained more clearly regarding why each variable is used in the analysis.

Author’s response: We thank the reviewer’s suggestion. The statistical analysis has been re-written. More details about how we form the logit. The levels of measurement for each study variable are described in the Methods section.

Result
-Regarding subscale analysis, the issue of multiple analyses present. The authors should show the p-values with Bonferroni correction.

Author’s response: We thank the reviewer’s suggestion. We understand the reviewer’s concerns about the increased risk of Type I error when conducting multiple tests or comparisons. However, our sample is relatively large, when compared to most of the clinical studies. Even in the separated analysis, each of the two subsamples has a sample size more than 2000. Plus. the statistical significance (p value = 0.022) for the latter study period (2004-2014) would still be significant with a Bonferroni correction (0.05/2 = 0.025 and 0.022 < 0.025).

-This section is not reader friendly and need subsections.

Author’s response: We thank the reviewer’s suggestion. We have slightly re-arranged and increased the spacing of this section to make more reader friendly. 

Discussion
With the same as the background, the authors should use paragraph writing for logical theory building. 

Author’s response: We thank the reviewer’s suggestion. The Discussion section has been re-written considerably. We adopted the paragraph writing style make our argument points clearer.

 This study should describe the limitation of sampling bias, the results' applicability to other settings, and the future investigation in the limitation part.

Author’s response: We thank the reviewer’s suggestion. We added more study limitations in this section. For example, we acknowledge that “not all the proxies for the HRS decedents knew all the details of end-of-life care planning pertaining to the deceased participants and about 20.3 percent of the 12,952 decedents had missing information on advance directives. It is possible that potential biases could be introduced due to the missing observations.”

In the conclusion or discussion, the study’s strengths should be focused on international readers.

 Author’s response: We thank the reviewer’s comments. We have added this as a study limitation. “This study focusses mostly on the birth cohorts of male HRS participants who had lived through a time when military conscription was instituted in the United States. Plus, the VHA health care system is supervised by the Department of Veteran Affairs, a US cabinet-level agency under the executive branch. These cohort-historic factors are culture- and country-specific, so the findings and implications of this study may not be applicable internationally.”

Conclusion
The conclusion section describes the result again. The authors should add a description of news regarding this research field.

Author’s response: We thank the reviewer’s comments. Actually, we did mention the overall trends in the use of advance directives in the US. “The percentage of older adults with a written advance directive has been rising since the Patient Self-determination Act was passed in 1990. Conducted on adults with various ranges of age, surveys on ACP activities indicated that some 23 to 54 percent Americans had a written advance directive in place and the rates were as high as 70 percent among older adults who had a terminal illness. In the current analysis, the percentages of having either a living will or a DPAHC among the decedents who died during the first-half study period were 47.8 percent for non-veterans and 54.2 percent for veterans.”

Reviewer 3 Report

Congratulations to the authors for dealing with a difficult, but at the same time very important topic.

The article can be improved in many aspects:

Introduction: it needs to be better clarified what is a veteran subject versus a non-veteran subject.Why 50 years old and not another age?

The bibliography is not current. I understand that on this subject there will not be much information, but one should try to look for primary documents and remove those that are more than 10 years old.

Plagiarism is 15.09%. Although it is acceptable, it could be lowered a little by rewriting the introduction.

The discussion could be improved. More articles should be included to contrast the results. It seems that the results are described again.

Author Response

Reviewer 3: Comments and Suggestions for Authors
Congratulations to the authors for dealing with a difficult, but at the same time very important topic.

The article can be improved in many aspects:

Introduction: it needs to be better clarified what is a veteran subject versus a non-veteran subject. Why 50 years old and not another age?

Author’s response: We thank the reviewer’s comments. This study is a secondary data analysis of a national representative survey, the Health and Retirement Study, addressing many important questions related to the challenging and opportunities of aging in America. The survey draws a multistage probability sample of older Americans aged 50 and over.

The bibliography is not current. I understand that on this subject there will not be much information, but one should try to look for primary documents and remove those that are more than 10 years old.

Author’s response: We thank the reviewer’s comments. It is true that many studies cited in this study dating back to the 2000s. However, they are cited because they are important studies from the top medical journals in the US, such as JAMA and New England Journal of Medicine, and they are still relevant currently.

Plagiarism is 15.09%. Although it is acceptable, it could be lowered a little by rewriting the introduction.

Author’s response: We thank the reviewer’s suggestion. The Introduction section has been re-written considerably.

The discussion could be improved. More articles should be included to contrast the results. It seems that the results are described again.

Author’s response: We thank the reviewer’s suggestion. The Discussion section also has been re-written considerably.

Round 2

Reviewer 2 Report

The manuscript has been considerably improved. I think that this paper is suited for inclusion in our journal.

Reviewer 3 Report

Congratulations on the update of the text. It has been improved. Thank you.